# Updates on the Proposed Botulinum Toxin A Mechanisms of Action in Orofacial Pain: A Review of Animal Studies

**DOI:** 10.3390/toxins17120567

**Published:** 2025-11-23

**Authors:** Jaime Fabillar Jr, Yumiko Yamamoto, Kazuyuki Koike, Daisuke Ikutame, Yoshizo Matsuka

**Affiliations:** 1Department of Stomatognathic Function and Occlusal Reconstruction, Graduate School of Biomedical Sciences, Tokushima University, Tokushima 770-8504, Japan or jmfabillar.swu@phinmaed.com (J.F.J.); koike.kazuyuki@tokushima-u.ac.jp (K.K.); c000030613@tokushima-u.ac.jp (D.I.); 2College of Dentistry, Southwestern University PHINMA, Cebu 6000, Philippines; 3Department of Bacteriology, Graduate School of Medicine, Dentistry and Pharmaceutical Sciences, Okayama University, Okayama 700-8558, Japan; yumiya@md.okayama-u.ac.jp

**Keywords:** botulinum toxin, orofacial pain, mechanism, animal models

## Abstract

Experimental animal models of orofacial pain have been instrumental in elucidating biological pathways underlying the antinociceptive effect of botulinum neurotoxin type A (BoNT/A). Although several mechanisms have been proposed to explain how BoNT/A relieves pain, the precise modes of action, particularly in the oral and maxillofacial areas, remain elusive. The purpose of this review was to synthesize and assess the latest proposed mechanisms of action through which BoNT/A attenuates orofacial pain in established animal models. A comprehensive search was conducted using the terms “botulinum neurotoxin,” “mechanism,” and “orofacial pain” or “trigeminal neuralgia.” Only animal studies involving the establishment of an orofacial pain model were selected. Additional relevant studies were identified through manual screening of cited references. Over the past five years, several animal pain models have been established to decipher the mechanisms underlying the BoNT/A-mediated antinociception in orofacial pain. The proposed mechanisms include retrograde transport, neuronal excitability regulation, neuropeptide inhibition, inflammatory modulation, and opioid system stimulation in both the peripheral and central nervous systems. Despite the insubstantial number of investigations and findings, BoNT/A exhibits multidimensional modulation of nociceptive responses and, therefore, remains a promising therapeutic agent for managing orofacial pain conditions, with animal studies consistently providing insights into the mechanism of its antinociceptive action.

## 1. Introduction

Orofacial pain has been defined by the International Association for the Study of Pain as a “frequent form of pain in the face and/or oral cavity” [1]. It comprises a diverse range of conditions that affect the dentoalveolar structures, masticatory muscles, temporomandibular joints, and neurovascular structures within the orofacial region [2], including neuropathic pain [3], inflammatory pain syndromes [4], burning mouth syndromes [5], temporomandibular disorders [6], and headache [7]. These pain conditions can substantially affect a person’s quality of life [8] and typically manifest as consistent discomfort, functional limitations, and psychological distress [9].

Botulinum toxin (BoNT) is primarily synthesized by the bacterium *Clostridium botulinum*, although similar toxin genes have also been identified in other *Clostridium* species, including *C. baratii*, *C. butyricum*, and *C. argentinense*. It structurally consists of light chain, heavy chain, and C-terminal receptor-binding domains [10]. It is a metalloprotease that blocks the acetylcholine release at the neuromuscular junction by cleaving the soluble N-ethylmaleimide-sensitive-factor attachment protein receptor (SNARE) complex involved in exocytosis [11], which consists of syntaxin, vesicle-associated membrane protein (VAMP), and synaptosomal-associated protein of 25 kDa (SNAP-25) [12]. Specifically, botulinum toxin type A (BoNT/A) targets the SNAP-25 protein [11]. BoNTs are traditionally used in cosmetic applications [13] and are routinely employed in the treatment of movement disorders such as cervical dystonia, spastic conditions, blepharospasm, and hyperhidrosis. The application of BoNTs for pain reduction has received growing recognition, prompting an expansion in their indications for conditions associated with chronic pain [14]. Accumulating evidence suggests that BoNT/A also exerts antinociceptive effects independent of its effect on muscles [15,16]. In clinical studies, BoNT/A has demonstrated potential for pain attenuation in conditions such as migraines [17], dystonia [18], and certain types of orofacial pain [3,19]. Although several mechanisms have been proposed to explain BoNT/A-mediated pain relief [10], the exact mechanisms—particularly in the oral and maxillofacial areas—remain unclear.

Experimental animal models of orofacial pain have played a crucial role in deciphering the biological pathways underlying the antinociceptive effects of BoNT/A [20,21,22,23,24,25,26]. These preclinical investigations have enabled controlled investigations of BoNT/A dynamics within both the peripheral and central nervous system milieu, as well as in the inflammatory environment. Established animal pain models provide valuable insights into the mechanisms through which BoNT/A attenuates pain—insights that are typically difficult to attain in human studies owing to methodological and ethical limitations.

The objective of the current review was to synthesize and assess the latest proposed mechanisms of action by which BoNT/A attenuates orofacial pain in established animal models. By summarizing the findings from heterogeneous experimental models, the review aims to highlight emerging mechanistic pathways, recognize knowledge gaps, and propose future research directions, clarifying the therapeutic potential of BoNT/A in the management of orofacial pain. A graphical representation of the synopsis is illustrated in Figure 1.

## 2. Methods

### 2.1. Study Selection

A comprehensive search in PubMed, including MEDLINE, was performed using the keywords “botulinum,” “mechanism,” and either “orofacial pain” or “trigeminal neuralgia.” This search yielded a total of 36 studies. An additional seven studies were identified through manual searching of the studies published within the last five years. The inclusion criteria comprised studies employing an orofacial pain model, through in vivo, ex vivo, or in vitro animal studies with the administration of BoNT/A, published in English between 2020 and 2025. Conversely, studies were excluded if they were review articles or case reports. Following removal of duplicates and screening for eligibility according to predefined inclusion and exclusion criteria, 13 studies were ultimately included in the analysis. A PRISMA flow diagram is illustrated in Figure 2. Relevant data were systematically extracted from each eligible study, encompassing study design, animal model specifications, intervention parameters (including BoNT/A dosage and route of administration), primary outcome measures, and significant mechanistic findings. Data extraction was independently carried out by the five authors, with subsequent cross-verification to ensure both accuracy and consistency. The synthesized data were qualitatively analyzed to elucidate mechanistic pathways, therapeutic implications, and existing knowledge gaps concerning the application of BoNT/A in orofacial pain models. To facilitate the discussion in this study, studies published prior to 2020 were also identified to corroborate the findings of the selected studies.

### 2.2. Risk of Bias Assessment

A comprehensive quality and risk-of-bias (RoB) assessment was conducted for all 13 included animal studies utilizing the SYRCLE Risk of Bias tool, an instrument derived from the Cochrane RoB tool and tailored for animal research. The SYRCLE tool examines ten domains of bias: (1) sequence generation, (2) baseline characteristics, (3) allocation concealment, (4) random housing, (5) blinding of personnel, (6) random outcome assessment, (7) blinding of outcome assessors, (8) incomplete outcome data, (9) selective reporting, and (10) other potential sources of bias. Each domain was independently evaluated by two reviewers, with categorizations of “low risk,” “high risk,” or “unclear risk.” Any disagreements were addressed through discussion to achieve consensus. The results of the assessment are presented in the Appendix A.

## 3. Proposed Mechanisms of BoNT/A in Orofacial Pain

Over the past five years, studies involving animal pain models have proposed numerous mechanisms by which BoNT/A elicits antinociception in orofacial pain conditions. These proposed mechanisms can typically be classified into transport mechanisms, neuronal excitability regulation, neuropeptide inhibition, inflammatory modulation, and opioid system stimulation in both the peripheral and central nervous systems.

### 3.1. Retrograde Axonal, Transsynaptic, and Hematogenous Transport Mechanisms

Beyond the peripheral site of administration, BoNT/A demonstrates retrograde axonal, transsynaptic, and hematogenous transport mechanisms in both the peripheral and central nervous systems. Nemanić et al. reported that BoNT/A can traverse the synapse toward the second-order neurons in the trigeminal nucleus caudalis (TNC), where it modulates central sensitization [27]. The authors demonstrated that unilateral subcutaneous administration of BoNT/A into the whisker pad area induced immunoreactivity of the cleaved SNAP-25 (cl-SNAP-25) in the bilateral trigeminal ganglia (TG), as well as in the bilateral TNC. Immunohistochemical analysis of cl-SNAP-25 successfully established the distribution of BoNT/A. The detection of cl-SNAP-25 within the TG and TNC on both the toxin-injected and contralateral sides implies that the peripherally administered toxin is transported retrogradely and transsynaptically in the central nervous system. These findings are consistent with those of Muñoz-Lora et al. [28], who reported the immunoreactivity of cl-SNAP-25 in the TNC following peripheral toxin administration directly into the temporomandibular joint (TMJ) of a TMJ pain model established by persistent immunogenic hypersensitivity (PIH). Moreover, Waskitho et al. reported the localization of BoNT/A (BoNT/A-Hc labeled with Alexa Fluor-488) in bilateral TG after unilateral subcutaneous administration into the whisker pad [29]. Their results suggested that the peripherally administered toxin was taken up by the axon terminals and subsequently transported retrogradely toward the neuronal soma in the bilateral TG. Additionally, the authors reported that approximately 18% of the subcutaneously administered toxin could be transported into the bloodstream, suggesting that the distribution of BoNT/A beyond the injection site may be mediated by its systemic transport. Although these investigations indicate peripheral, central, and systemic translocations, the overall evidentiary strength remains moderate due to a predominant reliance on immunohistochemistry, which lacks quantitative validation of functional toxin activity within peripheral and central neurons. A graphical representation is shown in Figure 3.

### 3.2. Regulation of Ion Channels, and Neuronal and Glial Excitation

BoNT/A regulates sensory neuronal channels, leading to reduced neuronal excitation. Moore et al. reported that in vitro toxin application suppressed inflammatory soup-induced sensitization of cultured trigeminal sensory neurons [30]. In treated neurons, cell surface expression of transient receptor potential (TRP) vanilloid subtype 1 (TRPV1) and TRP ankyrin 1 (TRPA1) was downregulated, and calcium influx was reduced (Figure 4A), suggesting disturbances in the trafficking of these channels to the plasma membrane. TRPV1 and TRPA1 are critical mediators in the pathophysiology of trigeminal pain [31]. Additionally, BoNT/A was shown to disrupt the trafficking of TRPV1 and TRPA1, resulting in inhibition of mechanical nociception [32]. Similarly, an ex vivo study by Kim et al. reported downregulation of TRPV1, TRPA1, and TRP canonical 1 (TRPC1) in the TG neurons, following the injection of BoNT/A into the TMJ of a forced mouth opening (FMO)-induced TMD mouse model [33] (Figure 4B).

Melo-Carillo et al. established a rat migraine model using cortical spreading depression (CSD) [34]. CSD-induced pain activates and sensitizes high-threshold and wide-dynamic range (WDR) dura-sensitive neurons in the spinal trigeminal nucleus. Extracranial administration of the toxin to the lambdoid and sagittal sutures inhibited the activation and sensitization of WDR neurons. The authors suggested that this inhibition could be attributed to the preferential inhibitory effects of the toxin on unmyelinated C-fibers. However, BoNT/A exerts only a limited inhibitory effect on the thinly myelinated Aδ-fibers, which could explain its inability to prevent activation of high-threshold neurons.

In a modified trigeminal neuralgia model established by Chen et al., chronic constriction injury to the distal infraorbital nerve led to the activation of glial cells in the TNC, as indicated by the upregulation of toll-like receptor 2 (TLR2), a marker of microglial activation [35]. This upregulation was suppressed by the administration of BoNT/A into the whisker pad area, as indicated by the reduced expression of TLR2 and CD11b (microglial markers), suggesting the inhibition of microglial cell activation.

Muñoz-Lora et al. successfully established an antigen-induced arthritic TMJ pain model that triggered microglial activation, as evidenced by elevated levels of the glia-neuron modulators Cathepsin S (CatS) and fractalkine (FKN), thereby activating the P2X purinoceptor 7 (P2X7)/CatS/FKN microglia-activated pathway [36]. Administration of BoNT/A to the TMJ downregulated protein levels of P2X7, CatS, and FKN. These findings suggest that the BoNT/A-mediated antinociceptive action is potentially associated with the modulation of the P2X7/CatS/FKN microglia-activated pathway.

Recently, Muñoz-Lora et al. established a TMJ pain model based on PIH [28]. Direct administration of BoNT/A into the TMJ revealed a substantial reduction in both neuronal and glial activation, as evidenced by the downregulation of the activation markers c-fos and GFAP, respectively.

These results highlight a shared mechanism whereby BoNT/A diminishes both peripheral and central excitability by modulating neuronal and glial activity. However, variations in the extent of these effects indicate that the efficacy of BoNT/A is modulated by neuroinflammatory factors specific to each experimental model, as well as by the distribution patterns of its target receptors.

### 3.3. Inhibition of Neurotransmitter Release

Following administration, BoNT/A enters neurons and cleaves SNAP-25, which is part of the SNARE complex [10]. Cleavage of SNAP-25 consequently inhibits the exocytosis of neurotransmitters such as acetylcholine [37]. Nemanić et al. reported that in the formalin-induced inflammatory orofacial pain model, BoNT/A administration induced immunoreactivity of cl-SNAP-25 in the TG and TNC [27]. To confirm the inhibition of neurotransmitter release, behavioral investigations were performed, revealing reduced nociception following BoNT/A administration.

In an in vitro toxin application study, Moore et al. reported similar findings regarding the cleavage of SNAP-25, preventing SNARE-mediated synaptic vesicle exocytosis in neurons and inhibiting neuropeptide release [30]. In particular, the authors reported marked attenuation of calcitonin gene-related peptide (CGRP) release mediated by TRPV1 and TRPA1. Likewise, Muñoz-Lora et al. reported that BoNT/A treatment did not increase the expression of CGRP [28]. In a migraine model established by Reducha et al., the administration of BoNT/A increased cleavage of SNAP-25 in the TG, along with a significant decrease in CGRP release in a hemi-skull model [38] (Figure 5).

Kim et al., in their ex vivo study, suggested that BoNT/A can potentially suppress glutamate release as it showed a decrease in the expression of glutamate-transporting protein (VGLUT2) in the TG neurons, after its injection into the TMJ of a forced mouth opening (FMO)-induced TMD mouse model [33].

The degree of neurotransmitter suppression reported across studies varies, likely attributable to differences in toxin preparation, route of administration (intra-TMJ versus subcutaneous), and the timing of assessments. Despite these methodological variations, this mechanism stands as one of the most consistently supported findings across diverse animal models.

### 3.4. Modulation of Neuroinflammation

More recently, studies have focused on the attenuation of neuroinflammation. For example, Shen et al. demonstrated that subcutaneous administration of BoNT/A into the frontal and temporal regions in an animal migraine model deactivated the nucleotide-binding oligomerization domain-like receptor protein 3 (NLRP3) inflammasome and downregulated interleukin (IL)-1β expression [39]. IL-1β is a well-recognized inflammatory cytokine [40], while NLRP3 inflammasome is an immune protein that activates caspase-1, which, in turn, cleaves the inactive form of IL-1β (pro-IL-1β) into its active form [41]. Both IL-1β and NLRP3 play crucial roles in the pathogenesis of migraine [39].

Similarly, in a lysophosphatidic acid (LPA)-induced trigeminal neuralgia model, a subcutaneous injection of BoNT/A into the whisker pad downregulated the expression of NLRP3, and well-known inflammatory cytokines IL-1β, IL-18, and tumor necrosis factor (TNF)-α [42].

In a trigeminal neuralgia model established by trigeminal nerve root compression, BoNT/A administration reduced levels of inflammatory cytokines in TG, specifically IL-1β, IL-6, and TNF-α [43] (Figure 6). In addition to inflammatory cytokines, the authors detected a substantial reduction in hypoxia-inducible factor (HIF)-1α, thereby suggesting that the antinociceptive effect of BoNT/A may involve regulation of an HIF-1α-related cytokine pathway.

Chen et al. reported downregulated expression of IL-1β, IL-6, and TNF-α in the central nervous system, particularly in the TNC following peripheral administration of BoNT/A [35]. Additionally, the expression of TLR2 was downregulated, suggesting that the antinociceptive action of BoNT/A may involve the TLR2-mediated cytokine pathway.

Manuel Muñoz-Lora et al. similarly reported that peripheral administration of BoNT/A led to downregulation of inflammatory cytokines, particularly TNF-α and IL-1β, in an antigen-induced arthritic TMJ pain model, in addition to central glial deactivation [36].

While the directionality of the observed effects is consistently reported, the robustness of the supporting evidence is variable. Differences in experimental inflammation models, such as LPA administration, nerve compression, or CFA injection, impede direct comparisons across studies. Nonetheless, despite these methodological constraints, the anti-inflammatory mechanism is robustly supported across a range of orofacial pain models.

### 3.5. Stimulation of the Opioid System

Lyu et al. reported that administration of BoNT/A increased the expression of nociceptin/orphanin FQ (N/OFQ) [44] (Figure 7), the fourth member of the opioid system. To elucidate the role of N/OFQ in BoNT/A-mediated antinociception, the authors administered UFP-101, an N/OFQ receptor antagonist, which suppressed the antinociceptive effect of BoNT/A, as evidenced by the greater nociceptive response in the BoNT/A + UFP-101 group compared with that in the BoNT/A + saline group. These findings suggest that the antinociceptive effect of BoNT/A could be mediated by the stimulation of N/OFQ expression.

This finding uncovers a potential new mechanism whereby BoNT/A influences endogenous opioid modulation; however, validation in independent experimental models has yet to be achieved. As such, the current evidence is preliminary and highlights the need for additional studies incorporating receptor antagonists or knockout models to definitively determine causality.

## 4. Orofacial Animal Pain Models Used to Assess BoNT/A Antinociception

Recent investigations between 2020 and 2025 have established various animal models to simulate pain conditions in the orofacial region. A summary of these animal models and the corresponding BoNT/A-mediated effects is presented in Table 1.

## 5. Discussion

Consolidated findings from recent animal studies have highlighted the diverse mechanisms through which BoNT/A modulates orofacial pain. Initially recognized for its muscle relaxant effects mediated by inhibiting the release of acetylcholine neurotransmitters at the neuromuscular junctions [13], subsequent investigations revealed that BoNT/A prevents the release of substance P, CGRP, somatostatin, serotonin, adenosine triphosphate, and bradykinin, all of which contribute to the sensitization of sensory nociceptors [45], acting on both neuronal and non-neuronal targets. Through its transport beyond the injection site, BoNT/A can influence both peripheral and central sensitization processes by altering the expression of ion channels, suppressing the release of neurotransmitters, regulating inflammatory mediators, and modulating opioid system dynamics. These mechanistic actions may occur independently of each other as they engage specific cellular targets. For instance, the cleaving SNAP-25 occurs predominantly in cholinergic neurons, while the downregulation of NLRP3 and IL-1β by BoNT/A is mediated primarily by immune and glial cells rather than neurons.

Retrograde axonal transport has been consistently documented [21,25,26], suggesting that the peripheral administration of BoNT-A can influence central pain pathways. It has been demonstrated by the detection of cleaved SNAP-25 or fluorescently labeled BoNT-A within TG and TNC after its peripheral administration via subcutaneous [27] or TMJ [28] injection, indicating active transport along sensory axons. Transsynaptic transport has been hypothesized based on studies demonstrating the presence of toxins or cleaved substrates in second-order neurons, in the absence of direct peripheral uptake. However, the supporting evidence for this mechanism remains predominantly indirect. Waskitho et al. proposed that the bilateral localization of BoNT/A in the TG following a unilateral whisker pad injection could be attributed to cell-to-cell trafficking, and reported hematogenous transport under experimental systemic conditions, but it is considered less likely under therapeutic dosing [29]. In a follow-up study utilizing a fluorogold neuronal tracer, bilateral localization in the TG via retrograde transport after unilateral administration could be attributed to peripheral diffusion, midline axonal crossover, and systemic transport [46]. In contrast, Cho et al. initially suggested that the antinociceptive effect of BoNT/A is mediated via a neuronal tissue transport mechanism rather than the recently proposed systemic transport mechanism. The authors reported subcutaneous administration of BoNT/A into the hind leg, distal to the nerve injury induced by trigeminal nerve root compression, failed to produce an anti-allodynic effect [43].

Ion channels in the peripheral terminal of nociceptor sensory neurons are crucial for generating sensory signals in response to noxious stimuli. By altering action potential generation and propagation, axonal conduction, and neurotransmitter release, these channels drive neuronal excitation and ultimately result in pain [47,48]. The TRP channel remains one of the most widely investigated channels in pain. Under mechanical hyperalgesia conditions, the expression of TRP melastatin 3 and TRPV4 [20], as well as TRPV1 and TRPA1 [32], was upregulated, and this upregulation was suppressed following BoNT/A administration. Similarly, recent studies have demonstrated that BoNT/A downregulated the expressions of TRPV1, TRPA1, and TRPC1 [30,33]. Noxious stimuli generate and propagate action potentials, which are conducted along the axon, leading to neuronal excitation. Glial cells, which are non-neuronal cells surrounding neurons, normally found in the nervous system, are also activated in response to nerve injury [49]. Recent animal studies have shown that BoNT/A interferes with the activation of neurons and glial cells by inhibiting the activation and sensitization of WDR neurons [34]. This effect is evidenced by the downregulated expression of the neuronal activation marker, c-fos [28]; reduced expression of glial activation markers TLR2, CD11b [35], and GFAP [28]; and decreased levels of protein of glia-neuron modulators, CatS and FKN [36].

It is widely recognized that the antinociceptive effects of BoNT/A are partly mediated by the inhibition of acetylcholine exocytosis at neuronal synaptic terminals [43]. Recent animal studies have demonstrated that treatment with BoNT/A can substantially attenuate the release of CGRP [28,30,38] and glutamate [33], indicating its ability to prevent SNARE-mediated synaptic vesicle exocytosis. Additionally, BoNT/A administration reduced nociception [27], confirming the inhibition of the exocytosis of pain-related neurotransmitters, as evidenced by immunohistochemical detection of cleaved SNAP-25 in both the peripheral and central areas [38]. Supporting these findings on inhibited neurotransmitter release, previous studies revealed that pretreatment with BoNT/A reduced the release of FM4-64, a membrane uptake marker, suggesting a decrease in excessive neurotransmitter release [24]. Regarding the antinociceptive mechanisms of BoNT/A, Aoki proposed that its beneficial effects on pain may be mediated through inhibition of neuropeptide release [50]. Consistently, Omoto et al. suggested that the release of neurotransmitters in the sensory ganglia plays an important role in the transmission of pain signals, as their study demonstrated that the direct administration of BoNT/A into the sensory ganglia alleviated painful behavior [51]. Similarly, intradermal administration of BoNT/A into the whisker pad has been shown to attenuate behavioral responses to thermal hyperalgesia [52].

Given that pain conditions are associated with inflammatory responses, it is crucial to understand inflammatory profiles for treating pain syndromes [4]. Several studies have explored inflammatory cytokines under pain conditions and whether BoNT/A can modulate pain experiences by mediating cytokine pathways. Recent investigations have focused on key inflammatory cytokines—such as IL-1β, IL-6, IL-18, and TNF-α—that are implicated in pain [35,36,39,42,43]. These inflammatory mediators were found to be markedly elevated in various orofacial pain models, and their levels are substantially reduced following BoNT/A administration. Treatment with BoNT/A also downregulated the expression of an immune protein that activates caspase-1, NLRP3 inflammasome [39].

The opioid system has been implicated in the antinociceptive effects of BoNT/A [26,53,54]. In a recent study, an orofacial nociception animal model was established through orthodontic tooth movement by closed-coil spring ligation of the teeth, and the administration of BoNT/A into the periodontal ligament induced the expression of N/OFQ, the fourth member of the opioid system [44]. Blocking N/OFQ suppressed the antinociceptive effect of BoNT/A, suggesting that this opioid plays a notable role in mediating the antinociceptive effect of BoNT/A.

Collectively, BoNT/A inhibits the transmission of nociceptive signals across the trigeminal system, affecting both the peripheral and central pathways. This supports its effects beyond muscular structures and underscores its potential as a therapeutic agent for treating orofacial pain conditions. Mechanistic insights derived from animal studies offer a foundation for understanding the potential of BoNT/A in alleviating orofacial pain within clinical contexts. However, the translation of these findings to patient care is inherently complex. Differences in BoNT/A formulation, dosing regimens, injection sites, and individual variability in pain pathophysiology may substantially influence therapeutic outcomes. Additionally, the extent of BoNT/A retrograde axonal transport and its subsequent impact on central pain pathways in humans has yet to be fully elucidated. Moreover, pain in humans typically involves complex biopsychosocial attributes that cannot be fully replicated in animal models. Thus, future clinical research integrating molecular and neuroimaging biomarkers is required to substantiate these preclinical mechanisms in human populations. Advancing knowledge in these translational domains is essential to bridge the gap between experimental models and clinical application, ultimately refining the utility of BoNT/A as a targeted intervention for orofacial pain.

Although recent studies on BoNT/A in animal models offer promising insights, their interpretability is hindered by methodological inconsistencies. Variations in dosing, injection parameters, assessment timing, and species selection greatly affect the reported antinociceptive outcomes. The prevalent use of male subjects also introduces a sex bias, limiting the applicability of findings, especially given the influence of hormonal and immune variables. Furthermore, most research focuses on short-term effects, leaving the long-term impacts and potential adaptive changes from repeated BoNT/A exposure insufficiently characterized. Addressing these methodological shortcomings is essential to improve experimental robustness and the translational relevance of BoNT/A research in orofacial pain.

## 6. Future Directions

To address the limitations of this study, the authors propose several directions for future investigation. Standardization of orofacial pain models is needed to enhance comparability across studies. Congruity in sensitization strategies, animal species, sex, and toxin preparation is essential for achieving reliable results. Additionally, further investigation of the central actions in higher-order neurons could provide novel insights, as current studies on the extent of toxin transport have been largely limited to second-order neurons. Moreover, future research may seek to elucidate which signaling pathways are causative rather than epiphenomenal, to investigate how timing and dosage influence retrograde transport, and to distinguish between the sensory and motor effects of BoNT/A. Finally, translational studies in human subjects through carefully tailored clinical trials can help bridge preclinical findings, thereby establishing the recommended route of administration, doses, and safety precautions for BoNT/A in the management of orofacial pain conditions.

## 7. Conclusions

Recent animal studies (2020 to 2025) have substantially expanded our understanding of BoNT/A-mediated mechanisms in attenuating orofacial pain conditions. When administered via various peripheral routes, BoNT/A impedes both peripheral and central sensitization following retrograde transport from the site of administration to the TG and TNC, where it regulates ion channels, inhibits neurotransmitter release, attenuates proinflammatory cytokines, and modulates the opioid system. These findings support the concept that BoNT/A is multidimensional in attuning nociceptive responses. However, a translational gap exists between animal models of pain and clinical conditions. Therefore, more robust studies are needed to bridge this gap and validate the clinical applications of these mechanisms. Nonetheless, BoNT/A remains a promising therapeutic agent for managing orofacial pain, and animal studies have provided insights into the mechanism of its antinociceptive action.

## Figures and Tables

**Figure 1 toxins-17-00567-f001:**
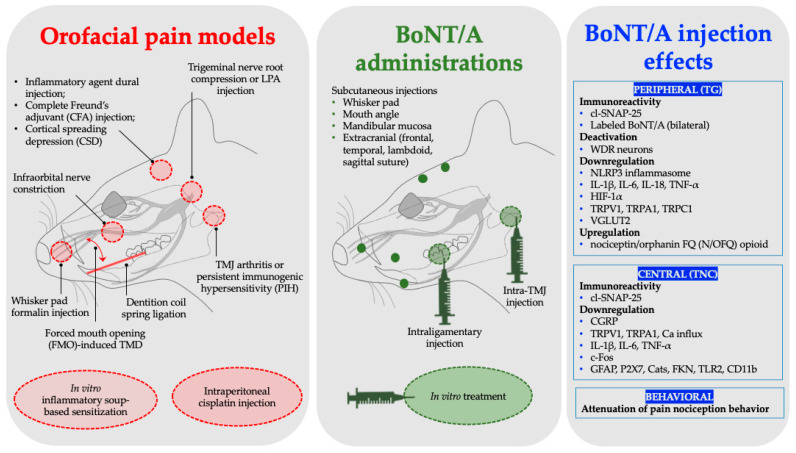
Consolidated orofacial pain models, BoNT/A administrations, and corresponding results. LPA, lysophosphatidic acid; TMD, temporomandibular joint disorders; TG, trigeminal ganglion; TMJ, temporomandibular joint; TNC, trigeminal nucleus caudalis; IL, interleukin; WDR, wide dynamic range.

**Figure 2 toxins-17-00567-f002:**
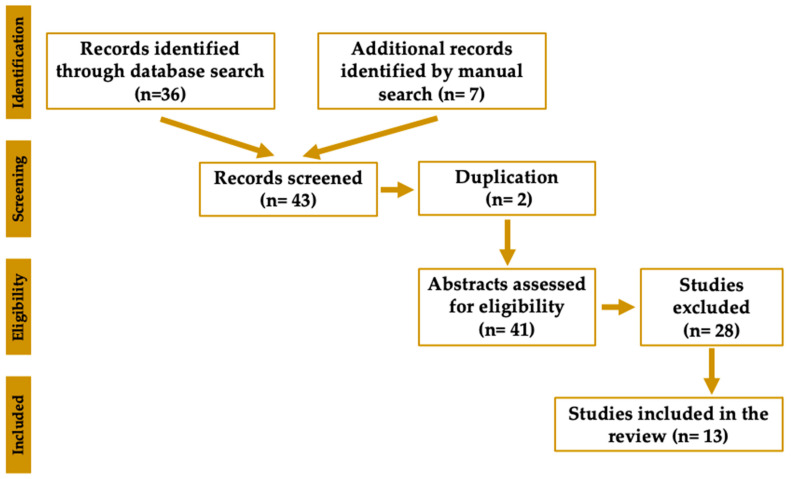
PRISMA flow diagram.

**Figure 3 toxins-17-00567-f003:**
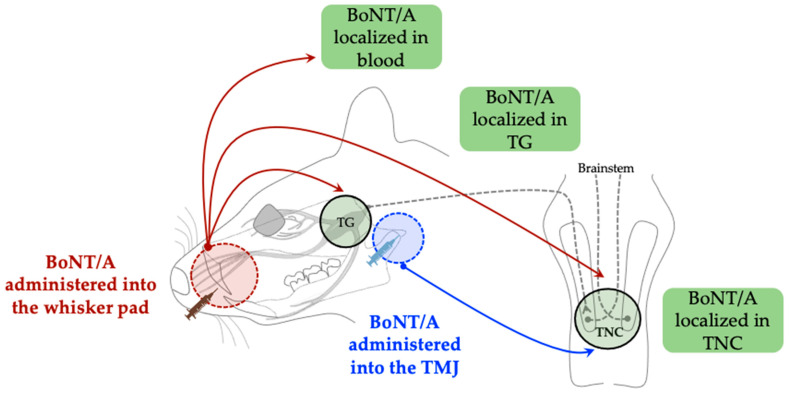
Peripherally administered BoNT/A shows retrograde axonal, transsynaptic, and hematogenous transport. Dashed circles denote the peripheral sites of BoNT/A administration. Red arrows depict the transport pathway following whisker pad injection, while the blue arrow indicates the transport pathway after TMJ injection. TG, trigeminal ganglion; TMJ, temporomandibular joint; TNC, trigeminal nucleus caudalis.

**Figure 4 toxins-17-00567-f004:**
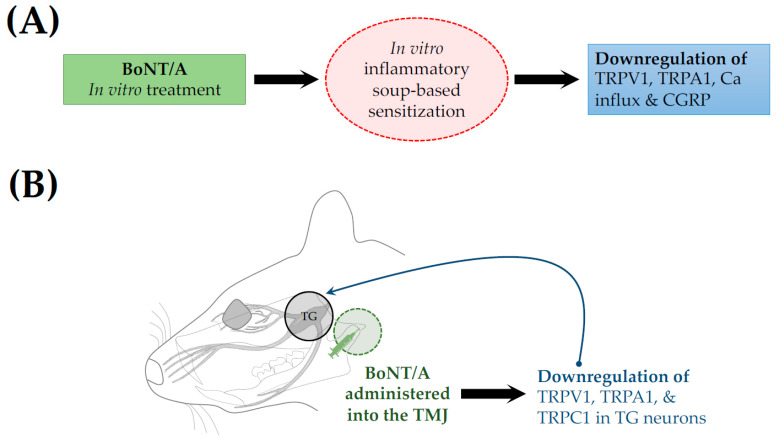
(**A**) In vitro application of BoNT/A shows downregulation of inflammatory cytokines and neuropeptides. (**B**) Forced mouth opening-induced TMD mouse model shows attenuation from BoNT/A administration. TRPV1, A1, and C1, transient receptor potential vanilloid, ankyrin, and canonical subtype 1; CGRP, calcitonin gene-related peptide.

**Figure 5 toxins-17-00567-f005:**
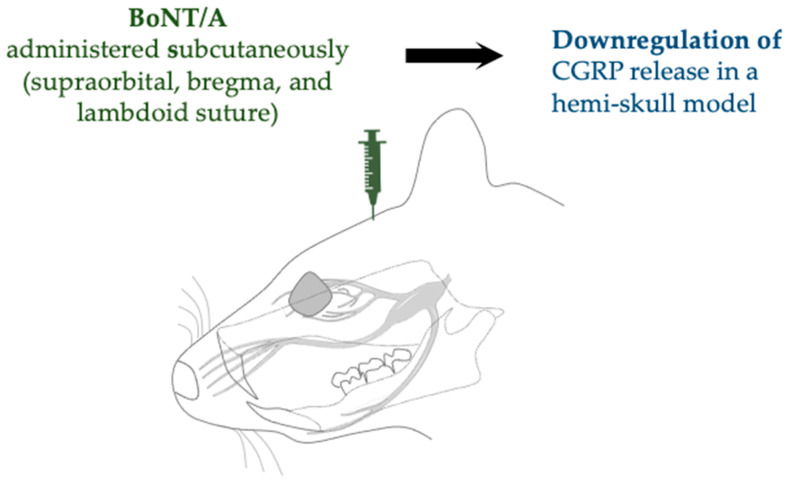
Migraine model by complete Freund’s adjuvant (CFA) model of inflammation administered with BoNT/A shows downregulation neuropeptides. CGRP, calcitonin gene-related peptide.

**Figure 6 toxins-17-00567-f006:**
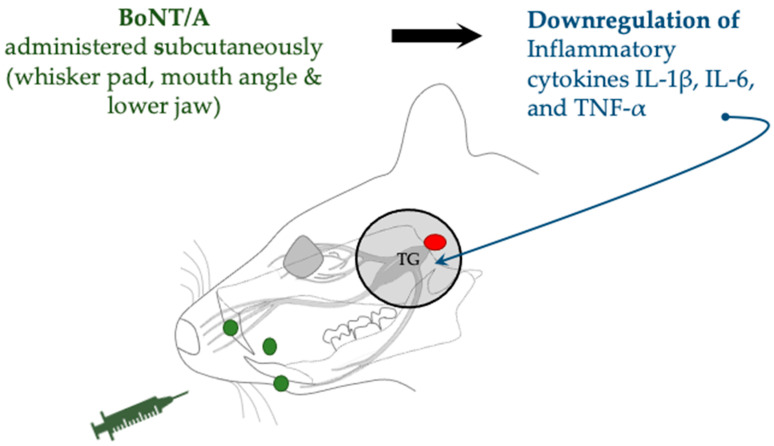
Trigeminal neuralgia model by trigeminal nerve root compression (red circle), administered with BoNT/A (green circles), shows downregulation of inflammatory cytokines. IL, interleukin; TNF, tumor necrosis factor.

**Figure 7 toxins-17-00567-f007:**
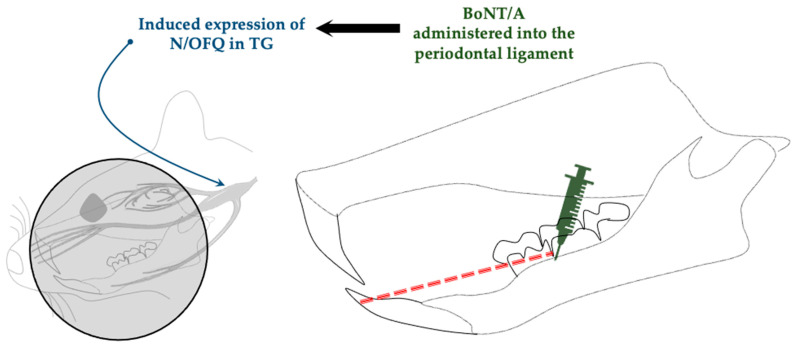
Orofacial nociception model by Orthodontic tooth movement through closed-coil spring ligation (red dashed line) shows activation of N/OFQ in TG (blue arrow) following intraligamentary administration of BoNT/A. N/OFQ, nociceptin/Orphanin FQ; TG, trigeminal ganglion.

**Table 1 toxins-17-00567-t001:** Established animal models for orofacial pain conditions and the corresponding BoNT/A effect.

Orofacial Pain Models		Animal Species	BoNT/A Administration	BoNT/A Dosage Administered	BoNT/A Effect Key Findings	Representative Studies
Migraine	Inflammatory agent dural injection	Sprague Dawley (SD) rats (male)	Subcutaneous (frontal and temporal)	10 U/Kg	Synthesis inhibition of NLRP3 and IL-1β	Shen et al., 2025 [39]
	Complete Freund’s adjuvant (CFA) model of inflammation	SD rats (male)	Subcutaneous (supraorbital, bregma, and lambdoid suture)	10 µL of 125 U/mL	Increased cleavage of SNAP-25 and downregulated expression of CGRP	Reducha et al., 2024 [38]
	Cortical spreading depression (CSD)	SD rats (male and female)	Extracranial injections (lambdoid and sagittal suture)	5 U/5 μL	Inhibition of activation and sensitization of WDR neurons	Melo-Carrillo et al., 2023 [34]
Inflammatory pain	Formalin injection	Wistar rat (male)	Subcutaneous (whisker pad)	7 U/Kg	Peripheral and central immunoreactivity of cl-SNAP-25	Nemanić et al., 2024 [27]
	Inflammatory soup-based sensitization model (in vitro)	C57BL/6J mice (males and females)	In vitro treatment	2.75 pM [50 U/mL; 2.5 ng/mL]	Reduction in TRPV1 and TRPA1 surface expression, decreased calcium influx, and inhibition of CGRP release	Moore et al., 2023 [30]
Temporomandibular joint pain	Persistent immunogenic hypersensitivity (PIH)	SD rats (male)	TMJ injection	7 & 14 U/Kg	cl-SNAP-25 immunoreactivity and neuronal and glial deactivation, inhibition of CGRP release	Muñoz-Lora et al., 2022 [28]
	Arthritis (Antigen-induced)	Wistar rats (male)	TMJ injection	7 U/Kg	Downregulated expression of P2X7, CatS, FKN, TNF-α, and IL-1β	Muñoz-Lora et al., 2020 [36]
	Forced mouth opening (FMO)-induced TMD	C57BL/6 & Pirt-GCaMP3 mice (male)	TMJ injection	20 µL [0.5 or 1 U]	Down regulated expression of vesicular glutamate transporter 2 (VGLUT2) protein, and TRPV1, TRPA1, and TRPC1	Kim et al., 2025 [33]
Trigeminal neuralgia	LPA trigeminal nerve root injection	SD rats (male)	Subcutaneous (whisker pad)	3 U/Kg	Downregulated expression of NLRP3, IL-1β, IL-18, and TNF-α	Park et al., 2025 [42]
	Trigeminal nerve root compression	SD rats (male)	Subcutaneous (whisker pad, mouth angle, and lower jaw)	1 or 3 U/kg	Expression downregulation of HIF-1α and IL-1β, IL-6, and TNF-α	Cho et al., 2022 [43]
	Distal infraorbital nerve-chronic constriction injury (dION-CCI)	C57BL/6 mice (male)	Subcutaneous (whisker pad)	0.18 U	Downregulated expression of TLR2, CD11b, IL-1β, IL-6, and TNF-α	Chen et al., 2021 [35]
	Infraorbital nerve constriction (IONC)	SD rats (male)	Subcutaneous (whisker pad)	10 MLD in100 µL saline	BoNT/A in the bilateral TG	Waskitho et al., 2021 [29]
Chemotherapy-induced neuropathic pain	Cisplatin injection	SD rats (male)	Subcutaneous (whisker pad)	10 MLD in100 µL saline	Behavioral pain attenuation	
Orthodontic movement- induced pain model	Dentition coil spring ligation	SD rats (male)	Intraligamentary injection (periodontal ligament)	1 U/6 µL	Promotes expression of nociceptin/orphanin FQ (N/OFQ)	Lyu et al., 2020 [44]

## Data Availability

No new data were created or analyzed in this study.

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
