# Peer review of "Updates on the Proposed Botulinum Toxin A Mechanisms of Action in Orofacial Pain: A Review of Animal Studies"

_toxins, 2025, doi:10.3390/toxins17120567_

Round 1
Reviewer 1 Report
Comments and Suggestions for Authors
Overall, the study is well-organized and well-written. I have a few comments regarding each section.
Introduction
The current definition of OFP does not reflect the complexity of these conditions, and needs to be developed by referencing other sources such as ICOP 2020.
Please clearly state the aim of the study in a concise way.
Method
Please correct ….PubMed includes MEDLINE, they are not separate databases.
It is unclear what criteria were used to limit the search. For example, was the search restricted by language?
Please define inclusion and exclusion criteria precisely.
The authors mention, the published studies during 2020—2025, but what types and designs?
How was “orofacial pain model” defined in the studies?
Were only BoNT/A studies included, or were other botulinum toxin types considered but then excluded to focus on the aim of the study?
The screening process needs to be completed by number of reviewers, if the studies screened independently, how were duplicate studies handled, and was any inter-reviewer agreement measured (kappa value)?
How data were extracted?
Please provide a PRISMA flow diagram.
Results and discussion
Please describe SNAP-25 and ion channels and glial cells, to make it easier for readers to understand the mechanism.
Generally, in the results section, the authors mainly describe the findings from various studies but do not analyze whether the results are conflicting or controversial findings. There is no information about how strong or reliable the evidence is, nor are the limitations of each study discussed including differences in methods or animal models. Another important issue is the clinical relevance of these mechanisms—how they translate to patient treatment.
By adding these missing parts, the discussion needs to be modified as well.
Author Response
REVIEWER 1:
Overall, the study is well-organized and well-written. I have a few comments regarding each section.
Introduction
The current definition of OFP does not reflect the complexity of these conditions, and needs to be developed by referencing other sources such as ICOP 2020.
Response: The authors agree that the definition of OFP in the manuscript, based on the definition by IASP, does not reflect the complexity of the conditions mentioned. Hence, we have the succeeding statement: “It comprises a diverse range of conditions that affect the dentoalveolar structures, masticatory muscles, temporomandibular joints, and neurovascular structures within the orofacial region [2]”, referencing ICOP 2020. We have highlighted this statement in the revised manuscript (Please see the Introduction section, lines 34-36).
Please clearly state the aim of the study in a concise way.
Response: We have revised and stated concisely the aim of the study, highlighting the statement in the revised manuscript (Please see Introduction section, lines 69-72).
Method
Please correct ….PubMed includes MEDLINE; they are not separate databases.
Response: Thank you for pointing this out. We have rephrased the statement, highlighting it in the revised manuscript (Please see Methods section, line 79).
It is unclear what criteria were used to limit the search. For example, was the search restricted by language?
Please define inclusion and exclusion criteria precisely.
The authors mention, the published studies during 2020—2025, but what types and designs?
Response: Thank you for these questions. We have specifically indicated the inclusion and exclusion criteria, adding that the search is limited to those published in the English language, as well as the specific types and design of study included. (Please see Methods section, lines 82-86).
How was “orofacial pain model” defined in the studies?
Response: Thank you for this question. The term “orofacial pain model” was not necessarily the term used in the studies analyzed. It is the term we decided to use to generally categorize the simulation of orofacial pain established in the studies.
Were only BoNT/A studies included, or were other botulinum toxin types considered but then excluded to focus on the aim of the study?
Response: Thank you for this question. To specifically focus on the aim of our study, we included only BoNT/A studies as specifically mentioned in the inclusion criteria. (Please see Methods section, line 84).
The screening process needs to be completed by number of reviewers, if the studies screened independently, how were duplicate studies handled, and was any inter-reviewer agreement measured (kappa value)?
How data were extracted?
Response: Thank you for this comment. We have now added a description of how the data were screened independently by the authors and extracted from the included studies. Specifically, we clarify that data were extracted manually using predefined parameters (study design, model, intervention, outcomes, and key findings) and cross-checked for accuracy. (Please see Methods section, lines 88-95)
Please provide a PRISMA flow diagram.
Response: Thank you for this relevant suggestion. A PRISMA flow diagram has been provided and is included in the revised manuscript, cited (please see line 88) and indicated as Fig. 2. (please see lines 98-99)
Results and discussion
Please describe SNAP-25 and ion channels and glial cells, to make it easier for readers to understand the mechanism.
Response: Thank you for these insightful suggestions. In order for the readers to understand the mechanism, we have explained that it has been suggested that BoNT acts on the SNARE complex and that BoNT/A specifically targets the SNAP-25 by cleaving it (Please see Introduction section, lines 45-49). Additionally, ion channels have been described and explained in the revised manuscript (Please see Discussion section, lines 320-323). Moreover, a description of glial cells has been added (Please see Discussion section, lines 329-331).
Generally, in the results section, the authors mainly describe the findings from various studies but do not analyze whether the results are conflicting or controversial findings.
There is no information about how strong or reliable the evidence is, nor are the limitations of each study discussed including differences in methods or animal models.
Response: Thank you for this insightful comment. We agree that the previous version primarily summarized findings without sufficient critical evaluation. In the revised manuscript, we have expanded the Results section to include a comparative analysis of the studies, highlighting areas of agreement and disagreement among the findings. (Please see section 3.1, lines 130-134; section 3.2, lines 178-182; section 3.3, lines 209-213; section 3.4, lines 247-252; section 3.5, lines 267-271).
Another important issue is the clinical relevance of these mechanisms—how they translate to patient treatment.
By adding these missing parts, the discussion needs to be modified as well.
Response: We appreciate the reviewer’s insightful comment emphasizing the need to address the clinical relevance of the mechanisms described in animal models. In response, we have expanded the Discussion section to include a paragraph that connects the mechanistic findings from animal studies to their potential translation into clinical treatment for orofacial pain. (Please see Discussion section, lines 374-386).

Reviewer 2 Report
Comments and Suggestions for Authors
Title: Please complete the title mentioning mecanisms "of action". Abstract: The abstract is clear but please clarify if the proposed mechanisms were only suggested or established through full experimental validation? Introduction: Line 40: BoNTs are also produced by Clostridium butyricum, Clostridium baratii and other bacteria. Line 42: blocks not 'suppresses' Line 47: The sentence is misleading. The BoNTs are used in cosmetics but also in the clinic for muscular hyperactivity, please list the therapeutic applications: "BoNTs are routinely used to treat movement disorders such as cervical dystonia, spastic conditions, blepharospasm and hyperhydrosis. The use of BoNTs to reduce pain has gained increased recognition, giving rise to an increasing number of indications in disorders associated with chronic pain." Rasetti-Escargueil et al. Toxins 2024 Methods: Line 69: Please provide the inclusion and exclusion criteria set to select the studies. Figure 1 would be more explanative in two separated panels: panel A for nociception and panel B for the different mecanisms with their respective localisations Section 3.1: a figure would be very informative specifically to show the trajectories of BoNT within the neuronal structures because the topic is complex. Please also clarify if the dose aministered during the studies were in the same range as clinical injections. It is not possible to extrapolate if the dose are much higher than in the clinic. Lines 138-142: In the studies by Chen et al, did the authors also confirm the transport of BoNT using immunomarkers? Line 161: acetylcholine is not a pain mediator, it is preferentially involved in neuromuscular force transmission but not in pain transmission. Section3.4: please clarify if the inactivation of NLRP3 and IL-1 beta is mediated by or independent from SNAP-25 cleavage? In Table 1 please indicate the toxin dose, this is essential to compare with the clinic and between studies. In the discussion, please summarize first the main findings of the review. In addition, please indicate how the SNAP-25 cleavage is involved in the different mechanisms described or if the mechanisms are independent. Sincerely yours
Author Response
REVIEWER 2:
Title: Please complete the title mentioning mecanisms "of action".
Response: Thank you for this suggestion. We added the suggested words to the title accordingly (Please see lines 2-3).
Abstract: The abstract is clear but please clarify if the proposed mechanisms were only suggested or established through full experimental validation?
Response: Thank you for this comment. The proposed mechanisms have been suggested based on their respective findings following experimental investigations.
Introduction: Line 40: BoNTs are also produced by Clostridium butyricum, Clostridium baratii and other bacteria.
Response: Thank you for pointing this out. We agree that while BoNTs are primarily produced by Clostridium botulinum, similar toxin genes have also been identified in other Clostridium species, including C. baratii (BoNT/F), C. butyricum (BoNT/E), and C. argentinense (BoNT/G), highlighting the genetic diversity and horizontal gene transfer among clostridial species. We have added this statement in the revised manuscript (Please see the Introduction section, lines 42-43).
Line 42: blocks not 'suppresses'
Response: Thank you for pointing this out. We have changed the term accordingly (Please see the Introduction section, line 45).
Line 47: The sentence is misleading. The BoNTs are used in cosmetics but also in the clinic for muscular hyperactivity, please list the therapeutic applications: "BoNTs are routinely used to treat movement disorders such as cervical dystonia, spastic conditions, blepharospasm and hyperhydrosis. The use of BoNTs to reduce pain has gained increased recognition, giving rise to an increasing number of indications in disorders associated with chronic pain." Rasetti-Escargueil et al. Toxins 2024
Response: Thank you for this valuable comment and suggestion. Accordingly, we have added clinical therapeutic applications of BoNT, citing the study by Rasetti-Escargueil et al. (Please see the Introduction section, lines 50-54).
Methods:
Line 69: Please provide the inclusion and exclusion criteria set to select the studies.
Response: Thank you for this valuable suggestion. We have elaborated the Methods section by specifying the inclusion and exclusion criteria that have been set in selecting the studies (Please see Methods section, lines 82-85).
Figure 1 would be more explanative in two separated panels: panel A for nociception and panel B for the different mecanisms with their respective localisations
Response: Thank you for this valuable suggestion. We agree with the reviewer’s suggestion; hence, we revised Figure 1 to a concise and clearer version, separating panels for the pain models, BoNT injections, and the resultant effects (Please see lines 74-76).
Section 3.1: a figure would be very informative specifically to show the trajectories of BoNT within the neuronal structures because the topic is complex. Please also clarify if the dose aministered during the studies were in the same range as clinical injections. It is not possible to extrapolate if the dose are much higher than in the clinic.
Response: Thank you for this significant suggestion. We agree that a figure could provide a better understanding; hence, an illustration pertaining to the text in section 3.1 has been added to enhance understanding for the readers. This illustration is indicated as Figure 3 (Please see lines 135-137). Additionally, we feel the need to do the same to the succeeding sections, thus, in the revised manuscript we also added figures as representative illustrations for section 3.2, indicated as Figure 4 (Please see lines 183-186); section 3.3 indicated as Figure 5 (Please see lines 214-216); section 3.4 indicated as Figure 6 (Please see lines 253-255); and section 3.5 indicated as Figure 7 (Please see lines 272-274).
Lines 138-142: In the studies by Chen et al, did the authors also confirm the transport of BoNT using immunomarkers?
Response: Thank you for this question. In the study by Chen et al, they utilized immunomarkers for the glial cell activation and inflammatory cytokines, which resulted in their respective reduced expressions following the treatment of BoNT/A. Their use of immunomarkers has tested the BoNT’s mechanistic action in the modulation of glial cell activation and neuroinflammation, but not with its transport mechanism. They may have, however, indirectly implicated BoNT’s retrograde transport since it was administered into the whisker pad and resulted in some molecular changes in the TG.
Line 161: acetylcholine is not a pain mediator, it is preferentially involved in neuromuscular force transmission but not in pain transmission.
Response: Thank you for pointing this out. We agree that acetylcholine is not directly involved as a pain-related neurotransmitter. However, since acetylcholine is mechanistically linked to BoNT action through the inhibition of its release at cholinergic synapses, we have revised the text accordingly. To maintain accuracy, we have deleted the term “pain-related” and clarified that BoNT affects the release of neurotransmitters such as acetylcholine (Please see Section 3.3, line 192).
Section3.4: please clarify if the inactivation of NLRP3 and IL-1 beta is mediated by or independent from SNAP-25 cleavage?
Response: Thank you for this question. The inactivation of the NLRP3 inflammasome and IL-1β is mediated primarily by immune and glial cells rather than neurons. Conversely, SNAP-25 cleavage occurs predominantly in cholinergic neurons and constitutes the classical mechanism by which muscle relaxation is achieved via inhibition of acetylcholine release. Consequently, the suppression or downregulation of NLRP3 and IL-1β by BoNT/A may occur independently of SNAP-25 cleavage, as these effects likely result from the engagement of distinct cellular targets.
In Table 1 please indicate the toxin dose, this is essential to compare with the clinic and between studies.
Response: Thank you for this insightful suggestion. Accordingly, we have added a column in Table 1 for the toxin dose and concentration administered in each study (Please see Table 1, line 283).
In the discussion, please summarize first the main findings of the review. In addition, please indicate how the SNAP-25 cleavage is involved in the different mechanisms described or if the mechanisms are independent. Sincerely yours
Response: We appreciate the reviewer’s insightful suggestion. The main findings of the review have been summarized in the first paragraph of the Discussion section, highlighted in the revised manuscript. Additionally, we have added in the text the independence of the different mechanistic actions owing to their different cellular targets or mediators (Please see Discussion section, lines 287-300).

Reviewer 3 Report
Comments and Suggestions for Authors
1. The Methods section briefly mentions searching MEDLINE and PubMed with three keywords (“botulinum,” “mechanism,” “orofacial pain/trigeminal neuralgia”) but does not describe: (1) Whether Boolean operators were used systematically, (2) Any use of MeSH terms, (3) Screening flowchart (e.g., PRISMA) to document inclusion/exclusion steps, (4) Quality assessment of included studies (e.g., SYRCLE’s risk of bias tool for animal studies).
2. Only 6 studies from 2020–2025 were included, plus 7 from reference screening. This is a narrow base for a mechanistic synthesis and raises concerns about selection bias.
3. The paper summarizes mechanisms (retrograde transport, ion channels, inflammation, neuropeptides, opioid system) but does not critically compare or contrast: (1) Dose–response variations, (2) Differences in animal models or administration routes, (3) Conflicting or inconsistent findings between studies.
4. Mechanisms are presented in silos (3.1–3.5) but lack a unifying framework that connects peripheral and central effects in a temporal or hierarchical sequence. A figure illustrating this cascade would strengthen the narrative.
5. While the Discussion acknowledges “variability in models,” it does not: (1) Specify which methodological variables (e.g., dose, site of injection, timing, species differences) most affect interpretation, (2) Address lack of female animal data (sex bias is common in pain research), (3) Highlight the absence of long-term or chronic exposure data in most studies.
6. Table 1 summarizes models but omits important mechanistic details: (1) BoNT/A dose or concentration, (2) Time between administration and measurement, (3) Specific markers evaluated (e.g., c-Fos, cytokines, receptor changes), (4) Whether effects were short-term or sustained.
7. Figure 1 is overly schematic. It does not visualize mechanistic interactions or temporal flow. A more mechanistically informative figure (e.g., showing transport from periphery to CNS with molecular targets) would enhance clarity.
8. The “Future Directions” section calls for standardization but doesn’t propose specific research questions, such as: (1) Which signaling pathways are causative vs epiphenomenal? (2) How timing/dose affects retrograde transport, (3) How to differentiate sensory vs motor effects of BoNT/A.
9. Terminology occasionally lacks precision — e.g., “retrograde axonal, transsynaptic, and hematogenous transport” are lumped without clear distinction of experimental evidence supporting each.
10. No statement about exclusion of non-English studies.
11. Abstract overstates “multidimensional modulation” without acknowledging the limited evidence base.
Author Response
REVIEWER 3:
- The Methods section briefly mentions searching MEDLINE and PubMed with three keywords (“botulinum,” “mechanism,” “orofacial pain/trigeminal neuralgia”) but does not describe: (1) Whether Boolean operators were used systematically, (2) Any use of MeSH terms, (3) Screening flowchart (e.g., PRISMA) to document inclusion/exclusion steps, (4) Quality assessment of included studies (e.g., SYRCLE’s risk of bias tool for animal studies).
Response: Thank you for the valuable comment. We have elaborated the Methods section by specifying the inclusion and exclusion criteria that have been set in selecting the studies, adding that the search is limited to those published in the English language, as well as the specific types and design of study included. In addition, we described how the data were screened independently by the authors and extracted from the included studies. Specifically, we clarify that data were extracted manually using predefined parameters (study design, model, intervention, outcomes, and key findings) and cross-checked for accuracy (Please see Methods section, lines 88-95). A PRISMA flow diagram has also been provided and is cited (please see line 88) in the revised manuscript, indicated as Fig. 2 (Please see lines 98-99).
- Only 6 studies from 2020–2025 were included, plus 7 from reference screening. This is a narrow base for a mechanistic synthesis and raises concerns about selection bias.
Response: Thank you for this comment. Our review is limited only to studies that utilized animal models in the investigation of the antinociceptive effect of BoNT/A treatment. To the best of our search strategy, we have only identified a few studies that align with the inclusion and exclusion criteria we have set. After careful identification and screening, we have determined 13 studies deemed suitable for our review.
- The paper summarizes mechanisms (retrograde transport, ion channels, inflammation, neuropeptides, opioid system) but does not critically compare or contrast: (1) Dose–response variations, (2) Differences in animal models or administration routes, (3) Conflicting or inconsistent findings between studies.
Response: Thank you for this significant observation. In the revised manuscript, we have added an analysis of each mechanistic action on the basis of the findings of the reviewed studies, including a comparative analysis of the studies, highlighting areas of agreement and disagreement among the findings. (Please see section 3.1, lines 130-134; section 3.2, lines 178-182; section 3.3, lines 209-213; section 3.4, lines 247-252; section 3.5, lines 267-271). Additionally, we have comparatively presented the differences in animal models, BoNT/A administration strategies, and their corresponding dosage or concentration in Table 1 (Please see line 283).
- Mechanisms are presented in silos (3.1–3.5) but lack a unifying framework that connects peripheral and central effects in a temporal or hierarchical sequence. A figure illustrating this cascade would strengthen the narrative.
Response: Thank you for this valuable suggestion. It was our intent in Figure 1 to present a consolidated illustration spatially showing the different animal models established, the BoNT/A administration, and the peripheral and central effects following its administration. Since it was considered overly schematic, we have revised Figure 1 in a less schematic way. Additionally, we have provided representative schematic illustrations of each mechanism (3.1-3.5) to visually aid in enhancing understanding among readers (Please see Figures 3-7).
- While the Discussion acknowledges “variability in models,” it does not: (1) Specify which methodological variables (e.g., dose, site of injection, timing, species differences) most affect interpretation, (2) Address lack of female animal data (sex bias is common in pain research), (3) Highlight the absence of long-term or chronic exposure data in most studies.
Response: Thank you for this valuable feedback. We have added text to the Discussion to address methodological variables. We now also note the limited representation of female subjects and note the predominance of short-term study outcomes and the scarcity of data on chronic or repeated BoNT/A exposure. These points clarify current research limitations and inform future investigative directions. (Please see Discussion section, lines 387-395).
- Table 1 summarizes models but omits important mechanistic details: (1) BoNT/A dose or concentration, (2) Time between administration and measurement, (3) Specific markers evaluated (e.g., c-Fos, cytokines, receptor changes), (4) Whether effects were short-term or sustained.
Response: Thank you for this insightful observation. Accordingly, we have revised Table 1, providing additional details such as the toxin dose and concentration administered in each study (Please see Table 1, line 283).
- Figure 1 is overly schematic. It does not visualize mechanistic interactions or temporal flow. A more mechanistically informative figure (e.g., showing transport from periphery to CNS with molecular targets) would enhance clarity.
Response: Thank you for this valuable observation. We agree with the reviewer’s comment, hence, we revised Figure 1 to a concise and clearer version, separating panels for the pain models, BoNT injections, and the resultant effects (Please see lines 74-76). Additionally, we feel the need to provide illustrations for each mechanistic action to enhance understanding for the readers. Thus, in the revised manuscript we also added figures as representative illustrations for section 3.1 indicated as Figure 3 (Please see line 135-137); section 3.2, indicated as Figure 4 (Please see line 183-186); section 3.3 indicated as Figure 5 (Please see line 214-216); section 3.4 indicated as Figure 6 (Please see line 253-255); and section 3.5 indicated as Figure 7 (Please see line 272-274).
- The “Future Directions” section calls for standardization but doesn’t propose specific research questions, such as: (1) Which signaling pathways are causative vs epiphenomenal? (2) How timing/dose affects retrograde transport, (3) How to differentiate sensory vs motor effects of BoNT/A.
Response: We appreciate the reviewer’s insight, and we agree that future direction should also propose relevant and significant questions that could potentially contribute to enhancing future investigations. Hence, we have added text in the Future Directions section as highlighted in the revised manuscript (Please see Future Directions section, lines 404-406).
- Terminology occasionally lacks precision — e.g., “retrograde axonal, transsynaptic, and hematogenous transport” are lumped without clear distinction of experimental evidence supporting each.
Response: Thank you for this comment. We appreciate the reviewer’s insightful comment. In the revised manuscript, we have clarified the distinctions among retrograde axonal (please see the Discussion section, lines 302-305), transsynaptic (lines 305-308), and hematogenous transport mechanisms (lines 311-312). Specifically, we now specify the experimental evidence supporting each mode of transport in the relevant animal studies.
- No statement about exclusion of non-English studies.
Response: Thank you for pointing this out. As part of the inclusion and exclusion criteria, we have added in the Methods section that the search is limited to those published in the English language, as well as the specific types and design of study included. (Please see Methods section, lines 82-86).
- Abstract overstates “multidimensional modulation” without acknowledging the limited evidence base.
Response: Thank you for pointing this out. We have revised the abstract, noting that there is a scarcity of referential investigations (Please see Abstract, lines 21-22).

Round 2
Reviewer 1 Report
Comments and Suggestions for Authors
Well done!
Author Response
Summary
We sincerely appreciate the constructive feedback provided by the reviewer, which has contributed significantly to the enhancement of our study. Thank you for your valuable insights. Please find the detailed responses below.
Point-by-Point Response to Comments and Suggestions for Authors
REVIEWER 1:
Well done!
Authors’ Response: We would like to express our gratitude to the reviewer for agreeing to assess our paper. We sincerely appreciate the time and effort taken to provide insightful revision suggestions, which will significantly enhance the quality of our study. Thank you for your valuable contributions.

Reviewer 3 Report
Comments and Suggestions for Authors
The revision addresses most concerns—including improved Methods detail, PRISMA flow, expanded mechanistic comparison, additional figures, and clearer discussion of limitations—but it still lacks a formal risk-of-bias or quality assessment of included studies, which remains a methodological weakness. Please perform any formal quality/risk of bias assessment for included 13 papers.
Author Response
Summary
We would like to express our sincere gratitude to the reviewer for the valuable time spent providing insightful comments and suggestions that have greatly enhanced the quality of this research. Your contributions are truly appreciated. Thank you very much. Please find the detailed responses below. The corresponding revisions/corrections are highlighted in the revised manuscript.
Point-by-Point Response to Comments and Suggestions for Authors
REVIEWER 3: The revision addresses most concerns—including improved Methods detail, PRISMA flow, expanded mechanistic comparison, additional figures, and clearer discussion of limitations—but it still lacks a formal risk-of-bias or quality assessment of included studies, which remains a methodological weakness. Please perform any formal quality/risk of bias assessment for included 13 papers.
Authors’ Response: We express our gratitude to the reviewer for emphasizing the significance of a formal quality and risk-of-bias (RoB) assessment. In response, we have conducted a comprehensive risk-of-bias evaluation for all 13 included animal studies using a validated and widely endorsed instrument, the SYRCLE Risk of Bias (SYRCLE-RoB) tool, specifically adapted from the Cochrane RoB tool for animal intervention research. Each study was assessed across the standard SYRCLE domains, including selection, performance, detection, attrition, reporting, and other biases. These evaluations are now detailed in the Methods section (Please see Methods section 2.2, pages 104-114). The findings are presented in the Supplementary material as a traffic light graph, along with an interpretative commentary. The authors believe that these enhancements could serve further to strengthen the methodological rigor and transparency of our review. Thank you very much.
